# Long-term exposure to polychlorinated biphenyl 126 induces liver fibrosis and upregulates miR-155 and miR-34a in C57BL/6 mice

**Fernanda Torres Quitete**[1☢], **Ananda Vitória Silva Teixeira**[1☢], **Thamara Cherem Peixoto**[1], **Bruna Cadete Martins**[1], **Geórgia Correa Atella**[2], **Angela de Castro Resende**[3], **Daniela de Barros Mucci**[1], **Fabiane Martins**[1,4], **Julio Beltrame Daleprane**[1] *

1 Department of Basic and Experimental Nutrition, Rio de Janeiro State University, Rio de Janeiro, Brazil, 2 Medical Biochemistry Institute, Federal University of Rio de Janeiro, Rio de Janeiro, Brazil, 3 Department of Pharmacology, Rio de Janeiro State University, Rio de Janeiro, Brazil, 4 Department of Morphology, Federal University of Rio Grande do Norte, Rio Grande do Norte, Brazil

☢ These authors contributed equally to this work.
* beltrame@uerj.br

**Data Availability Statement:** All relevant data are within the manuscript and its Supporting information files.

## Abstract

Environmental pollutants, including polychlorinated biphenyls (PCBs), act as endocrine disruptors and impair various physiological processes. PCB 126 is associated with steatohepatitis, fibrosis, cirrhosis, and other hepatic injuries. These disorders can be regulated by microRNAs (miRNAs). Therefore, this study aimed to investigate the role of miRNAs in non-alcoholic fatty liver disease associated with exposure to PCB 126. Adult male C57BL/6 mice were exposed to PCB 126 (5 μmol/kg of body weight) for 10 weeks. The PCB group showed lipid accumulation in the liver in the presence of macro- and microvesicular steatosis and fibrosis with increased inflammatory and profibrotic gene expression, consistent with non-alcoholic steatohepatitis (NASH). PCB exposure also upregulated miR-155 and miR-34a, which induce the expression of proinflammatory cytokines and inflammation in the liver and reduce the expression of peroxisome proliferator-activated receptor α, which, in turn, impairs lipid oxidation and hepatic steatosis. Therefore, the present study showed that PCB 126 induced NASH via potential mechanisms involving miR-155 and miR-34a, which may contribute to the development of new diagnostic markers and therapeutic strategies.

## Introduction

Environmental factors such as persistent organic pollutants have been highlighted as endocrine disruptors capable of interfering with physiological processes. Exposure to these pollutants, even at low doses, is associated with obesity and non-alcoholic fatty liver disease (NAFLD) [1–4]. Polychlorinated biphenyls (PCBs) are widely used in industrial processes, and despite their banned production, are still present in the environment because of their high thermodynamic stability and resistance to biodegradation [2, 5, 6]. PCBs, including their more

**Funding:** This research was supported by the Fundação Carlos Chagas Filho de Amparo à Pesquisa do Estado do Rio de Janeiro-FAPERJ (grant numbers E-26/211.193/202, E-26/201.234/2022, and E-26/210.332/2022) and the Coordenação de Aperfeiçoamento de Pessoal de Nível Superior-CAPES, 001.

**Competing interests:** The authors have declared that no competing interests exist.

toxic congener PCB 126, are present in the diet [7] and are associated with obesity, NAFLD, insulin resistance, diabetes, oxidative stress, and metabolic syndrome [1–3, 8–12]. Their mechanism of action involves the aryl hydrocarbon receptor (AhR), which regulates the transcription of several genes involved in NAFLD progression [2, 11]. These findings are concerning as the NHANES 2003–2004 showed that 100% of the adult population had detectable serum levels of PCBs [5]. Additionally, a 2010 study involving an American population estimated that the daily intake of PCBs was approximately 30 ng/day [13].

PCBs are associated with inflammation, accumulation of triglycerides in the liver, steatohepatitis, fibrosis, cirrhosis, and hepatocarcinoma [1–3, 8–12]. These disturbances are associated with various molecular markers, such as microRNAs (miRNAs), clarifying their participation in the onset and progression of various pathological processes, either as diagnostic molecular markers or potential therapeutic targets [14–17]. NAFLD is strongly associated with miR-155, miR-34a, and miR-122, as these miRNAs possibly regulate different processes of lipid metabolism and are involved in the inflammatory process characteristic of NAFLD [14–16, 18]. miR-122 downregulation and miR-34a upregulation are consistently found in individuals with NAFLD [16]. In contrast, miR-34a and miR-155 are associated with obesity, accompanied by the accumulation of triglycerides in the liver, as they target the mRNA of molecules such as sirtuin 1, which is reduced in individuals with NAFLD and acts by regulating energy homeostasis through the modulation of transcription factors [14–17].

Although several studies have shown an association between miRNAs and the onset of NAFLD [14–17], to date, no study has investigated the association between the expression patterns of these miRNAs and exposure to PCBs. Therefore, the present study aimed to investigate the participation of miRNAs in the development of NAFLD associated with exposure to PCB 126 to clarify the molecular mechanisms by which these disorders occur, providing candidate diagnostic markers and new therapeutic strategies.

## Material and methods

### Animals and experimental groups

Male C57BL/6 mice aged 3 months old were maintained on a 12 h/12 h dark/light cycle with controlled humidity (60 ± 10%) and temperature (21 ± 2˚C) and free access to food and water. This study was approved by the Animal Ethics Committee of the State University of Rio de Janeiro (protocol number CEUA/013/2019) in accordance with the ARRIVE guidelines and was performed in accordance with the National Research Council's Guide for the Care and Use of Laboratory Animals. Mice were randomly assigned to two groups (n = 10 for each group) according to treatment with PCB 126 (5 μmol/kg of body weight, diluted in corn oil) [19, 20] or vehicle (corn oil), which was administered through intragastric gavage biweekly at weeks 2, 4, 6, and 8 of the study (Fig 1A).

All animals received a normocaloric and normolipidemic standard diet (14%, 10%, and 76% of energy as protein, fat, and carbohydrates, respectively; total energy, 15 kJ/g). The diet was manufactured by PragSolucoes (Jaú, São Paulo, Brazil) and was in agreement with the recommendations of the American Institute of Nutrition (AIN 93M) [21].

Food intake and body mass were measured weekly. At the end of the experiment (10 weeks), the animals were fasted for 6 h and then deeply anesthetized with an intraperitoneal injection of sodium thiopental [60 mg/kg body mass, associated with 2% lidocaine (10 mg)] to withdraw blood using a heparinized syringe by cardiac puncture. Blood samples were centrifuged at 3000 × $g$ for 15 min at 4˚C to obtain the plasma. The samples were stored individually at −20˚C until the analytical assays. In addition, livers were carefully dissected, weighed, frozen

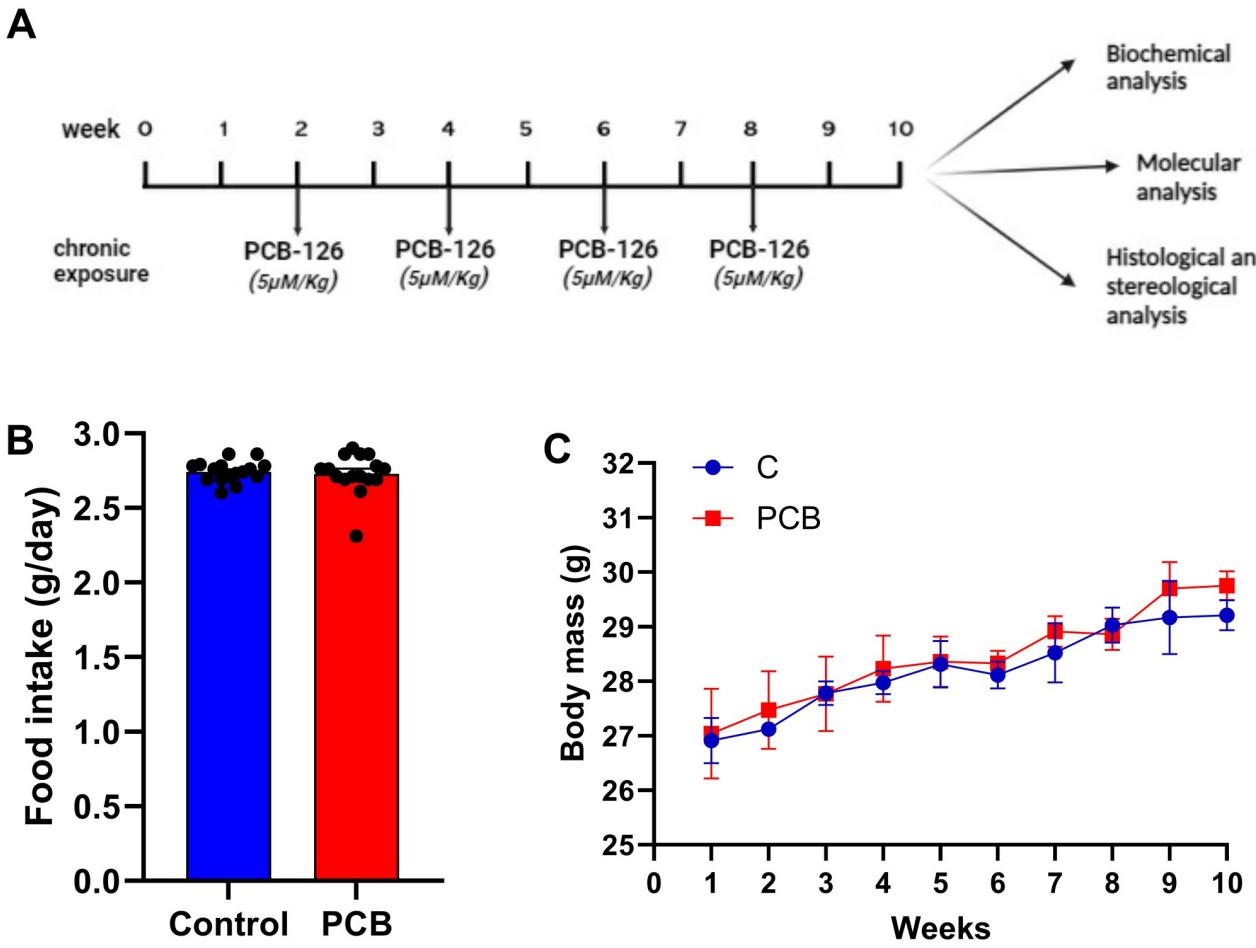

**Fig 1. Characterization of the experimental model.** (A) Overview of the experimental approach: C57BL/6 male mice were allocated to receive vehicle–control group or PCB 126 (5 μM/kg) biweekly for 10 weeks. At the end of 10 weeks, mice were euthanized and blood and liver were collected for further analyses. (B) Average food intake per animal per day. (C) Body mass during 10 weeks of experiment; values expressed as mean ± standard deviation (n = 10 per group).

in liquid nitrogen, and stored at −80°C or fixed in formalin for subsequent analyses and histology, respectively.

## Plasma analyses

Total cholesterol, triglyceride, aspartate aminotransferase (AST), alanine aminotransferase (ALT), and fasting glucose concentrations in plasma samples were evaluated using commercial colorimetric kits (Bioclin, Belo Horizonte, Brazil). Insulin (#EZRMI-13-K, Millipore, MO), interleukin 6 (IL-6; #BMS603-2, Invitrogen, CA, USA), and tumor necrosis factor-α (TNF-α; #88-7324-88 Invitrogen, CA, USA) levels in plasma were determined using enzyme-linked immunosorbent assay (ELISA) with commercially available kits.

## Histological and stereological analysis of the liver

After being fixed in formalin, hepatic tissue was placed in Paraplast Plus (Sigma-Aldrich, St Louis, MO, USA). Subsequently, 5-μm sections were placed on slides and stained with hematoxylin-eosin and Sirius red. The slides were analyzed and images were captured randomly

(JPG format, 36-bit color, 1360 × 1024 pixels) under a light microscope (Olympus BX51 with DP71 digital camera; Olympus Optical, Tokyo, Japan). The volume density (Vv) of hepatocytes was determined using the STEPanizer software version 1.8 via 16-point tests. The results were calculated by dividing the sum of the points found by the sum of the total points of the system and were expressed as percentages [22, 23]. Alternatively, to evaluate interstitial fibrosis, the slides were stained with Sirius red solution for 1 h and counterstained with hematoxylin [24, 25]. Furthermore, the extent of fibrosis was quantitatively measured and characterized as the proportion of tissue area positively stained with Picrosirius Red relative to the total liver tissue area, as described previously [22, 26].

## Determination of hepatic cholesterol, triglyceride, and hydroxyproline contents

Liver samples (50 mg) were homogenized in 1 mL isopropanol (Vetec, Rio de Janeiro, Brazil) and centrifuged (50 x $g$/10 min/4˚C). Total cholesterol and triglyceride levels in the supernatant were measured using a colorimetric method with a commercial kit (Bioclin, Belo Horizonte, Brazil). Hydroxyproline levels were quantified using a colorimetric method (catalog # MAK008; Sigma-Aldrich, St. Louis, MO, USA) and expressed as micrograms of hydroxyproline per milligram of liver tissue.

## Determination of liver fatty acid composition

Total lipids from the liver were extracted as described by Bligh and Dyer [24], with modifications. After incubation in a chloroform-methanol-water solution (2:1:0.8, v/v), 10 mg of liver sample was centrifuged (1500 × $g$ for 20 min at 4˚C). Chloroform was added to the supernatant. After centrifugation (1500 × $g$ for 20 min), the organic phase was removed and dried under a nitrogen stream. The extracted lipids were analyzed using thin-layer chromatography (TLC) for neutral lipids on a DC Silica gel 60 plate (Merck Millipore, HE, Germany) [27]. After the run, the plates were submerged for 10 s in Charring solution (3% $CuSO_4$ and 8% $H_3PO_4$ (v/v)), dried, and heated to 110˚C for 10 min. The TLC plates were analyzed using densitometry (Image Master software from Total Lab, Auckland, New Zealand). The samples were also analyzed using gas chromatography–mass spectrometry (GC–MS) [28]. Lipid samples were dissolved in 1% sulfuric acid in methanol. The GC-MS analysis was performed on a Shimadzu GCMS-QP 2010 Plus system using an HP Ultra 2 column (5% phenyl methylpolysiloxane; Agilent Technologies, 25 m × 0.20 mm × 0.33 μm). The injector temperature was set to 250˚C. The column temperature was programmed from 40 to 160˚C at 30˚C/min, 160–233˚C at 1˚C/min, 233–300˚C at 30˚C/min, and then held at 300˚C for 10 min. Electron ionization (EI-70 eV) was performed using a quadrupole mass analyzer operating at scans from 40 to 440 amu. The interface was set at 240˚C and the ion source at 240˚C. Lipid components were identified by comparing their mass spectra to those of the NIST05 MS library contained in the mass spectrometer. Retention indices were used to confirm the identity of the peaks in the chromatogram using the Supelco 37 Component FAME Mix certified reference material (Sigma-Aldrich, St. Louis, MO, USA).

## Total RNA and miRNA extraction and real-time reverse transcription polymerase chain reaction (RT-qPCR)

Total RNA and miRNA were extracted from the liver under RNase-free conditions using RNAzol RT (RN 190) reagent (Molecular Research Center, Cincinnati, OH, USA). Total RNA and miRNA were quantified using a NanoVue Plus Spectrophotometer (GE Healthcare,

Buckinghamshire, UK). cDNA was prepared from total RNA using a High-Capacity cDNA Reverse Transcription Kit (catalog #:4368814) (Applied Biosystems, Foster City, CA, USA) and from miRNA using a TaqMan MicroRNA Reverse Transcription Kit (catalog #:4366597) (Applied Biosystems, Foster City, CA, USA). The mRNA levels of *Ahr* (assay ID: Mm00478932_m1), *Mcp1* (assay ID: Mm00656886_m1), *Cxcl1* (assay ID: Mm04207460_m1), *Il-6* (assay ID: Mm00446190_m1), *Tnfa* (assay ID: Mm00443258_m1), *Tgfβ1* (assay ID: Mm03024091_m1), *Smad3* (assay ID: Mm03024086_m1), *Collagen1α* (assay ID: Mm01309565_m1), *Cxcl9* (assay ID: Mm00434946_m1), *Caspase3* (assay ID: Mm01195085_m1), *Sirt1* (assay ID: Mm01168521_m1), and *Ppara* (assay ID: Mm00440939_m1) and the miRNA levels of miR-155 (assay ID: 002571), miR-122 (assay ID: 002245), and miR-34a (assay ID: 00426) in the liver were determined. The mRNA and miRNA levels of the markers cited above were measured using TaqMan Fast Advanced Master Mix (catalog #4444963) (Applied Biosystems, Foster City, CA, USA) according to the manufacturer's instructions. RT-qPCR was performed in triplicate for each sample using an Applied Biosystems 7500 Fast Real-Time PCR System (Applied Biosystems, Foster City, CA, USA). Oligonucleotide primers and probes were prepared by Applied Biosystems (Foster City, CA, USA). Co-amplification of mouse *Gapdh* mRNA (assay ID: Mm99999915_g1) and U6 snRNA (assay ID: 001973), with various internal controls, was performed for all samples. The results were normalized to GAPDH mRNA and U6 snRNA levels using the $2^{-\Delta\Delta CT}$ method. This method can be used to calculate relative changes in gene expression, as determined by real-time quantitative PCR [29].

## Western blotting analysis

Hepatic protein extraction involved liver homogenization with buffer and protease inhibitors, followed by centrifugation [25]. Supernatants were collected and equal protein amounts were suspended in SDS-buffer, heated, and separated via SDS-PAGE. Electrophoresis used 30 μg protein aliquots, transferred to nitrocellulose membranes. Membranes were blocked with TBS-T solution containing 5% albumin and incubated overnight with cleaved caspase-3 (Cell Signaling, Massachusetts, USA), caspase 3, SMAD 6 and β-actin (Santa Cruz Biotechnology, Santa Cruz, CA, USA) antibody. After washing, membranes were incubated with secondary antibodies, washed again, and protein expression was detected using ECL kit (GE Healthcare, Madison, WI) and ChemiDoc Resolutions System. Quantitative analysis was performed using Image Pro Plus software v 7.01.

## Statistical analysis

The results are shown as the means ± standard deviation. GraphPad Prism 10.2 (GraphPad Software, La Jolla, CA, USA) was used for statistical analyses and graphics. All data were tested for normal distribution and homogeneity of variance using Bartlett's test. Experimental data were analyzed using Student's *t*-test, and differences were considered significant at $p < 0.05$.

## Results

### Food intake, body mass, and plasma analysis

No differences were observed in average daily food intake (Fig 1B) or body mass (Fig 1C) between the groups during the 10-week experimental period. The PCB group showed higher plasma triglyceride, glucose, and insulin levels (+68%, +68%, and +53%, respectively; $p < 0.001$; Table 1) than the C group. Plasma markers of hepatic function were also evaluated, and the PCB group showed increased levels of ALT and AST (4.9- and 1.7-fold increase,

**Table 1. Biochemical parameters in plasma.**

| Plasma analysis | C | PCB |
|---|---|---|
| ALT (U/L) | 9.6 ± 4.1 | 56.9*** ± 16.1 |
| AST (U/L) | 70.4 ± 2.9 | 191.8*** ± 3.1 |
| Cholesterol (mmol/L) | 1.45 ± 0.2 | 1.23* ± 0.2 |
| Triglycerides (mmol/L) | 0.37 ± 0.0 | 0.62*** ± 0.1 |
| Glucose (mmol/L) | 6.50 ± 1.2 | 10.9*** ± 1.2 |
| Insulin (pmol/L) | 93.3 ± 1.6 | 142.8*** ± 2.6 |
| IL-6 (pg/ml) | 45.4 ± 3.3 | 73.8*** ± 3.8 |
| TNF-α (pg/ml) | 14.0 ± 1.5 | 20.0*** ± 1.1 |

C–Control group; PCB–PCB 126 group.

Values expressed as mean ± SD (n = 10).

* $p$ value < 0.05;

** $p$ value < 0.01;

***$p$ value < 0.001.

respectively; $p < 0.001$; Table 1) compared to the C group. In addition, the PCB group showed higher levels of the proinflammatory cytokines IL-6 and TNF-α (+63% and +43%, respectively; $p < 0.001$, Table 1) than the C group. Surprisingly, the plasma cholesterol concentration was lower in the PCB group than in the C group (-15%; $p < 0.05$, Table 1).

## Liver analysis

Histological and stereological analyses were performed on the liver samples, and the stained sections showed microvesicular and macrovesicular steatosis of small and large lipid droplets dispersed in the livers of animals exposed to PCB (Fig 2A). Vv [steatosis, liver] hepatic steatosis was higher in the PCB group than in the C group (3.5-fold increase; $p < 0.001$, Fig 2B). Consistent with this result, the PCB group showed higher hepatic triglyceride (+98%; $p < 0.001$, Fig 2C) and cholesterol (+40%; $p < 0.001$, Fig 2D) contents than the C group. PCB serves as a classical ligand for the aryl hydrocarbon receptor (*AhR*), and the activation of this transcription factor is associated with the exacerbation of hepatic steatosis. As anticipated, its expression was elevated in the PCB group compared to that in the control group (2.5-fold increase, $p < 0.0001$; Fig 2E). In addition to macro- and microsteatosis in the liver, inflammatory infiltrates were observed on histological slides. The expression levels of *Mcp1* (3-fold increase; $p < 0.0001$, Fig 2F) and *Cxcl1* (2-fold increase; $p < 0.001$, Fig 2G) were higher in the PCB group than in the control group. Consistent with systemic inflammation findings, the hepatic expression of *Il-6* (0.5-fold increase, $p < 0.01$, Fig 2H), *Tnf-α* (2-fold increase, $p < 0.01$, Fig 2I).

Furthermore, the PCB group exhibited fibrotic areas, suggesting potential progression from hepatic steatosis to non-alcoholic steatohepatitis (NASH) (Fig 3A). To confirm the presence of fibrosis, we quantified the Sirius red staining area, and the PCB group showed a 4.5-fold increase in Sirius red staining than the control group (Fig 3B, $p < 0.0001$). Hydroxyproline quantification was 6 times higher in the PCB group than in the control group (Fig 3C, $p < 0.0001$). Genes involved in the development of fibrosis, namely *Tgfβ1* (3.5-fold increase, $p < 0.0001$, Fig 3D), *Smad3* (4-fold increase, $p < 0.0001$, Fig 3E), *Collagen1α* (3.4-fold increase, $p < 0.0001$, Fig 3F), and *Cxcl9* (3-fold increase, $p < 0.0001$, Fig 3G), were also upregulated in the PCB group compared to the control group. Hepatic fibrosis, which may result from steatosis, is exacerbated by the death of hepatocytes. The PCB group exhibited increased *Caspase 3* gene expression (2-fold increase; Fig 3H) and increased cleaved caspase 3 protein expression

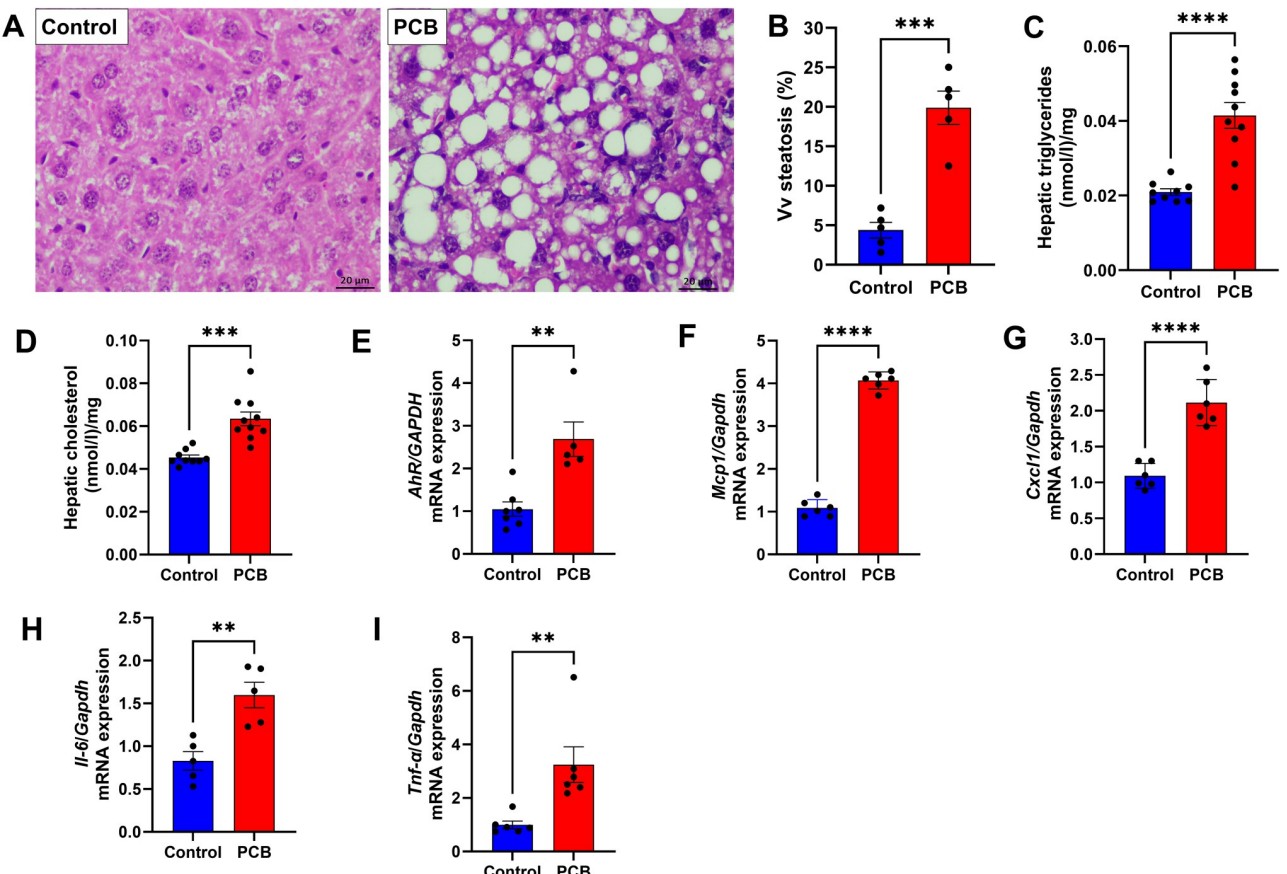

**Fig 2. Effects of PCB 126 on cellular, morphological, and molecular features of the liver after 10 weeks of exposure.** (A) Stereology of the liver; (B) Vv steatosis (%) of mouse liver; (C) hepatic triglyceride content; (D) hepatic cholesterol content; (E) aryl hydrocarbon receptor (AhR) gene expression in the liver; gene expression of inflammation-related genes: (F) *Mcp1*, (G) *Cxcl1*, (H) *Il-6* and (I) *Tnf-α*. C: Control group; PCB: PCB 126 group. Values are expressed as mean ± standard deviation (n = 10 per group for hepatic lipid content; n = 5 per group for histological and stereological analysis; n = 8 per group for RT-PCR). ** *p* value < 0.01; *** *p* value < 0.001; ****p value < 0.0001.

(Fig 3I and 3J; p<0.001). Protein level of SMAD 6 was up-regulated in PCB group (Fig 3K and 3L; p < 0.001).

The fatty acid profile was also evaluated in liver samples, and no differences were observed in the amounts of saturated, monounsaturated, and polyunsaturated fatty acids between groups or in the amounts of n-6 and n-3 essential fatty acids (Table 2).

In addition, SIRT1 and PPAR-α, biomarkers associated with steatosis development, were assessed in the liver. No differences were observed in *Sirt1* mRNA expression (Fig 4A), whereas the mRNA expression of *Ppara* decreased in the PCB group compared to the C group (-63%; *p* < 0.05, Fig 4B). Furthermore, miRNAs associated with NAFLD were evaluated in the liver, and no differences were observed in the expression of miR-122 (Fig 4C); however, the expression of miR-34a (1.7-fold-increase; *p* < 0.05, Fig 4D) and *miRNA-155* (1-fold increase, *p* < 0.001, Fig 2E) were elevated in animals exposed to PCB when compared with the C group.

## Discussion

Organic pollutants, particularly PCBs, have been identified as important factors that promote health problems. In this context, PCB 126 is associated with the development of obesity,

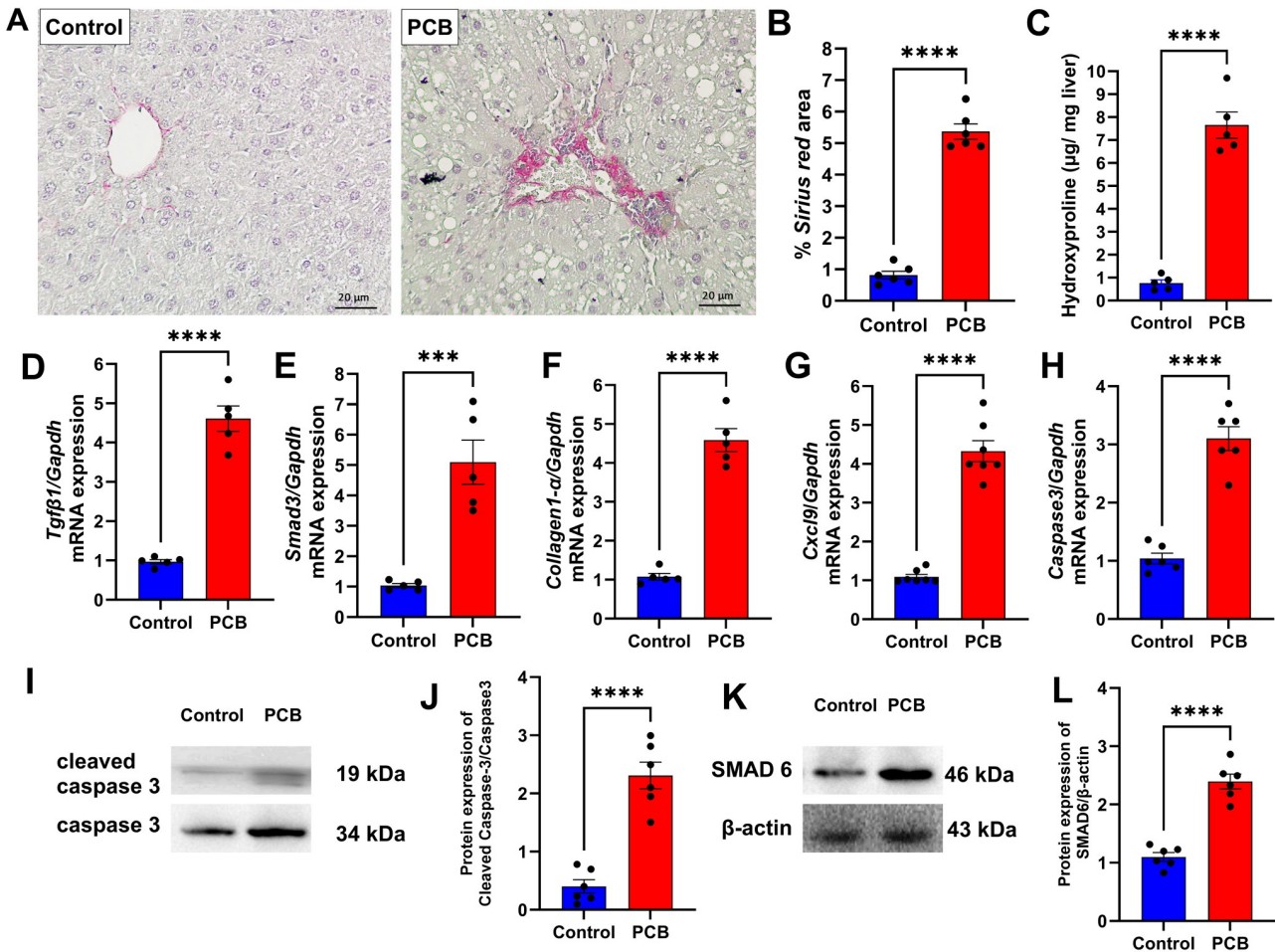

**Fig 3. Histological and molecular parameters associated with liver fibrosis after chronic PCB 126 exposure.** (A) Picrosirius red staining; (B) %
Picrosirius Red area; (C) hydroxyproline content in the liver; profibrogenic gene expression: (D) *Tgfβ1*, (E) *Smad3*, (F) *Collagen1α*, (G) *Cxcl9*, and cell
death-related gene: (H) *Caspase 3*. Protein Levels of cleaved caspase 3 (I and J) and SMAD 6 (K and L) were assed. C: Control group; PCB: PCB 126
group. Values are expressed as mean ± standard deviation (n = 8). ** p value < 0.01; *** p value < 0.001.

inflammatory processes, and NAFLD [1, 2, 4]. The identification of molecular mechanisms
and biomarkers that regulate the development of these disorders, such as miRNAs, is of great
importance to provide a better understanding of the participation of these compounds in the
current health panorama of the population and to guide public policies in the care of exposure
to these substances.

The present study evaluated the impact of chronic exposure to PCB 126 for 10 weeks in
mice, and, at the end of treatment, no differences were observed in food intake or body mass
between the groups. However, the PCB group showed altered biochemical parameters in
plasma, such as an increase in markers of hepatic function, ALT and AST, increased levels of
triglycerides, glucose, and insulin, and lower levels of cholesterol, indicating that, although this
group did not present an altered body composition, exposure to PCB 126 disturbed some met-
abolic processes. An altered profile of glucose metabolism with elevated plasma glucose con-
centrations was observed in mice exposed to different types of PCB, including PCB 126 [30,
31]. This hyperglycemic condition possibly results from insulin resistance in the liver, muscle,
and adipose tissue through mechanisms involving TNF-α which, at higher levels, impair the

**Table 2. Fatty acids profile in liver samples.**

| Fatty acids | C | PCB |
|---|---|---|
| Lauric acid (C12:0) | 0.326 | 0.144 |
| Tridecanoic acid (C13:0) | 0.000 | 0.000 |
| Myristic acid (C14:0) | 0.609 | 0.591 |
| Pentadecanoic acid (C15:0) | 0.190 | 0.145 |
| Palmitic acid (C16:0) | 17.998 | 17.298 |
| Margaric acid (C17:0) | 0.265 | 0.193 |
| Stearic acid (C18:0) | 6.582 | 6.704 |
| Arachidonic acid (C20:0) | 0.271 | 0.228 |
| Behenic acid (C22:0) | 0.211 | 0.190 |
| Tricosanoic acid (C23:0) | 0.000 | 0.069 |
| Lignoceric acid (C24:0) | 0.298 | 0.284 |
| Myristoleic acid (C14:1) | 0.000 | 0.044 |
| Palmitoleic acid (C16:1) | 3.234 | 4.330 |
| Cis-10-heptadecenoic acid (C17:1) | 0.201 | 0.156 |
| Oleic acid (C18:1n9c) | 21.117 | 19.585 |
| Elaidic acid (C18:1n9t) | 4.632 | 6.512 |
| Cis-11-eicosenoic acid (C20:1n9) | 1.220 | 1.229 |
| Erucate acid (C22:1n13) | 0.438 | 0.286 |
| Nervonic acid (C24:1n9) | 0.394 | 0.334 |
| Gamma-linolenic acid (C18:3n6) | 0.237 | 0.164 |
| Linoleic acid (C18:2n6c) | 16.964 | 21.517 |
| Arachidonic acid (C20:4n6) | 9.172 | 6.429 |
| cis-5.8.11.14.17-Eicosapentaenoic acid (C20:5n3) | 0.405 | 0.308 |
| Dihomo-alpha-linolenic acid (C20:3n6) | 1.826 | 2.323 |
| cis-11.14-eicosadienoic acid (C20:2) | 0.799 | 1.097 |
| cis-4,7,10,13,16-docosapentaenoic acid (C22:5n6) | 0.774 | 0.399 |
| cis-4.7.10.13.16.19-Docosahexaenoic acid (C22:6n3) | 9.442 | 6.632 |
| 7,10,13,16-Docosatetraenoic acid (C22:4n6) | 0.984 | 0.799 |
| Docosapentaenoic acid (C22:5n3) | 1.425 | 1.856 |
| Cis-13,16-docosadienoic acid (C22:2) | 0.000 | 0.144 |
| Saturated fatty acids (%) | 26.75 | 25.85 |
| Monounsaturated fatty acids (%) | 31.24 | 32.48 |
| Polyunsaturated fatty acids (%) | 42.03 | 41.67 |
| Total (%) | 100 | 100 |

C–Control group; PCB–PCB 126 group.

Values expressed as percentage (%).

insulin signaling cascade and glucose uptake by the GLUT4 transporter in insulin-dependent tissues [30, 32]. Although not the focus of the present study, these results are similar to those observed here, showing that animals exposed to PCB 126 present with impaired glycemic homeostasis.

In addition, the PCB group showed a proinflammatory profile in plasma, with elevated levels of IL-6 and TNF-α. Exposure to PCB 126 induces the expression of inflammatory cytokines, including TNFα and IL-1β, possibly promoting the polarization to the proinflammatory M1 phenotype in a human monocyte cell culture. In addition, an increase in monocyte chemoattractant protein-1 (MCP-1) in PCB 126-activated macrophages suggests the induction of

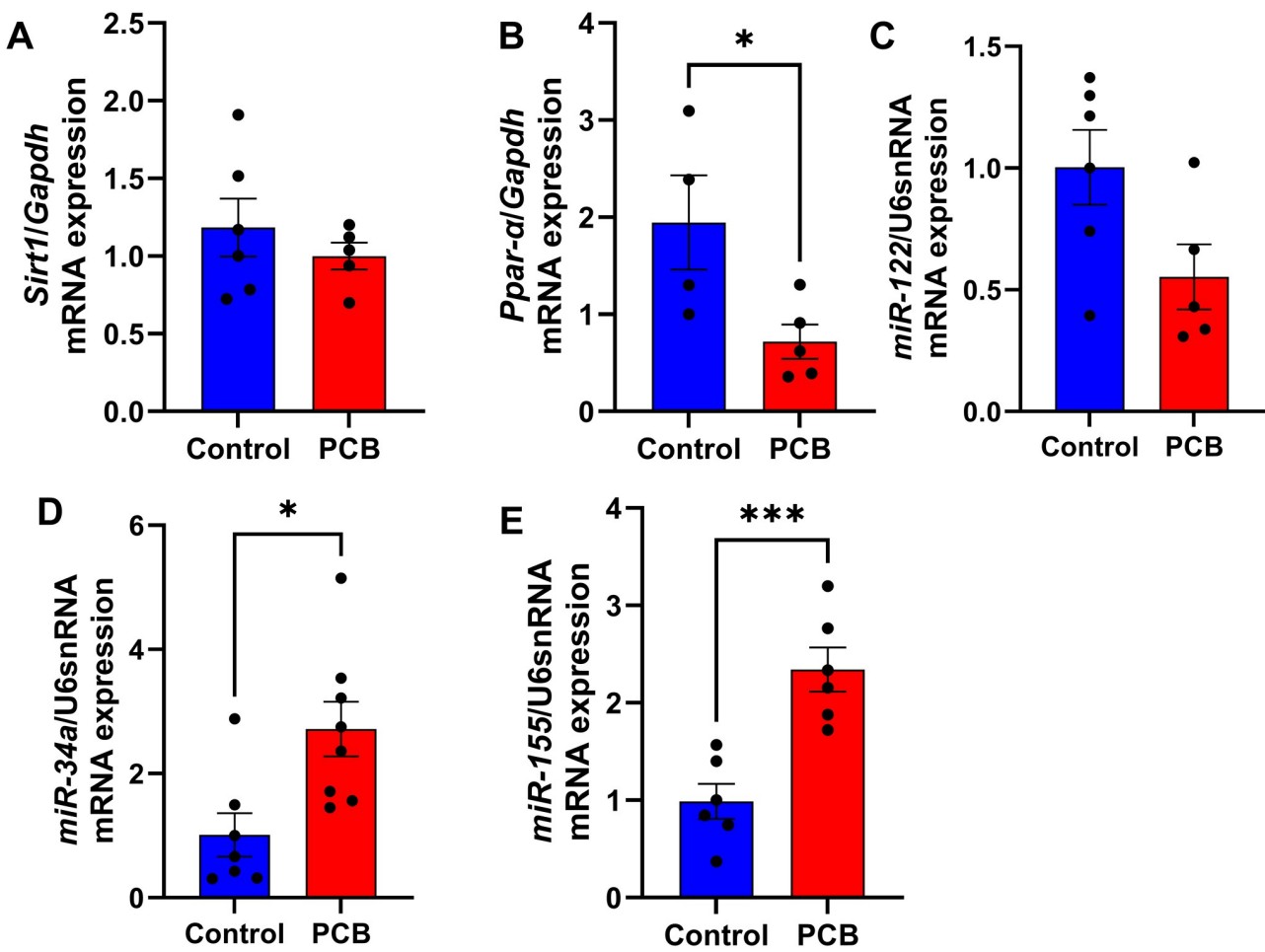

**Fig 4. Molecular parameters of lipid metabolism in the liver after chronic PCB 126 exposure.** (A) Sirtuin-1 (*Sirt1*) gene expression in the liver. (B) Peroxisome proliferator-activated receptor alpha (*Ppar-α*) gene expression in the liver. (C) *miR-122* expression in the liver (D) *miR-34a* expression in the liver. (E) *miR-155* expression in the liver. C: Control group; PCB: PCB 126 group. Values are expressed as mean ± standard deviation (n = 8). * *p* value < 0.05; *** *p* value <0.001.

chemokines regulating immune cell recruitment and infiltration of monocytes/macrophages [33]. Additionally, the gene expression of major chemokines for different immune cell infiltrations was higher in the livers of the PCB group, suggesting chemotaxis of the inflammatory cells.

To confirm that PCB 126 promotes NAFLD, liver samples were evaluated and the PCB group showed lipid accumulation in the presence of macro- and microvesicular steatosis and increased hepatic cholesterol and triglyceride contents. Consistent with steatohepatitis, the PCB group exhibited fibrosis and inflammatory infiltration of the liver. PCB 126 interferes with metabolic pathways in the liver and there is an association between this pollutant and the molecular development of NAFLD. Exposure to PCB 126 increases lipid accumulation in hepatocytes and triglyceride concentrations in the liver of rats through mechanisms involving microsomal triglyceride transfer protein and diacylglycerol O-acyltransferase 2 (DGAT-2), which are important molecules in the hepatic synthesis and export of triacylglycerides [1].

Administration of PCB 126 to mice resulted in notable histological damage and vacuolar degeneration in the liver, with the accumulation of lipid droplets, increased inflammation, and

collagen accumulation, showing liver fibrosis [34]. Furthermore, similar to the results of the present study, liver injury was confirmed when serum AST and ALT levels increased [34]. These results, which are in agreement with the data presented in this study, demonstrate that PCB 126 induces liver inflammation, fibrosis, and injury in mice.

All the effects described for PCB 126 appear to occur through the AhR, a receptor in the liver to which PCB binds and regulates the transcription of several genes involved in NAFLD development [2, 11]. In the present study, *Ahr* gene expression increased in the liver of the group exposed to PCB 126, suggesting that this pollutant may promote its effects through this receptor. PCB-126 is a potent endocrine disruptor that interferes with thyroid and steroid hormone receptors, affecting liver lipid metabolism. This exposure may increase cholesterol synthesis or hinder its elimination by altering liver metabolic pathways and enzyme activities like HMG-CoA reductase. PCB-126 also induces oxidative stress and inflammation in the liver, which can damage cells and trigger a defensive increase in cholesterol synthesis. Additionally, it impacts the function and expression of lipid transport proteins, such as LDL and HDL, affecting cholesterol transport and removal [35, 36].

Furthermore, the expression of *Tnfa* and *Il6* was evaluated in the liver, and PCB 126 was found to positively regulate the expression of these cytokines, showing that these animals presented increased inflammation in the liver tissue. This is consistent with the histological and stereological analyses, where lipid accumulation, inflammatory infiltration, and fibrosis were verified, showing the development of NASH. These results corroborate previous studies that showed that PCB 126 promotes an inflammatory state in the liver with an increase in proinflammatory cytokines such as TNF-α, IL-1β, IL-6, MCP-1, C-C Motif Chemokine Ligand 2 (Ccl2), Ccl3, and Ccl5, contributing to the development of liver injury [33, 34].

In the present study, genes involved in lipid metabolism, such as *Sirt1* and *Ppara*, were evaluated in the livers of the animals. SIRT1, a protein deacetylase dependent on NAD+, plays a pivotal role in maintaining the energy balance within the body. Specifically, hepatic SIRT1 governs lipid equilibrium by enhancing the function of PPARα, a nuclear receptor crucial for adapting to fasting and starvation. When SIRT1 is deficient in liver cells, it undermines PPARα activity, diminishing fatty acid oxidation. This disruption contributes to the onset of hepatic steatosis and inflammation, particularly in the context of a high-fat diet. [37]. In the present study, PCB 126 did not alter the expression of *Sirt1* but reduced the expression of *Ppara*. It is important to highlight that the present study evaluated the gene expression of *Sirt1*. Although no changes were observed in the PCB group, the hypothesis that the protein content or enzymatic activity may be reduced and promote the reduction of gene expression of *Ppara* cannot be ruled out. In this context, activation of PPARα leads to increased expression of several target genes involved in fatty acid oxidation, and lack of PPARα, in a mouse model, promotes an excessive accumulation of triglycerides in the liver [38, 39].

Accumulation of triglycerides in the liver tissue, inflammation, steatohepatitis, liver fibrosis, and cirrhosis are associated with microRNAs, suggesting their participation in the development and progression of NAFLD [14–17, 40]. The present study evaluated the expression of miRNAs that have already been associated with these disorders, such as miR-155, miR-122, and miR-34a [14–17], to clarify the possible contribution of these markers in regulating the changes observed in the livers of animals exposed to PCB 126. The mechanism by which PCB-126 (polychlorinated biphenyl-126) regulates miR-155 and miR-34a expression is closely tied to its interaction with the aryl hydrocarbon receptor (AhR) signaling pathway [41]. PCB-126 is a potent ligand for AhR, a transcription factor that is activated upon binding with various ligands, including environmental pollutants like dioxins and PCBs. When PCB-126 binds to AHR, it triggers the translocation of the AhR-ligand complex into the nucleus [20, 42]. Once inside the nucleus, the AhR-ligand complex can bind to specific DNA sequences known as

xenobiotic response elements (XREs) located in the promoter regions of various genes, including those encoding for miRNAs like miR-155 and miR-34a. This binding can either upregulate or downregulate the transcription of these miRNAs, depending on the context and the specific interactions at the gene promoters. miR-155 and miR-34a are microRNAs that play crucial roles in various biological processes, including inflammation, cell cycle regulation, and apoptosis [33, 36, 43].

The regulation of these miRNAs by AhR signaling can influence these processes, potentially leading to various biological effects. The alteration of miR-155 and miR-34a expression by PCB-126 through AhR signaling can have significant pathophysiological implications [3]. For example, dysregulation of these miRNAs may contribute to the development of diseases like cancer, where they are known to play roles in regulating apoptosis, proliferation, and the immune response. In summary, PCB-126 regulates miR-155 and miR-34a expression primarily through its interaction with AhR, leading to changes in gene transcription that can have broad biological and health implications. This pathway is a critical aspect of understanding the toxicological effects of PCBs and their impact on cellular and molecular processes [20, 43–45].

Liver expression of miR-155 increased in the PCB group, and this miRNA plays a role in the regulation of Kupffer cells (KCs) and is involved in the inflammatory processes in NAFLD [40]. miR-155 is highly expressed in total liver, hepatocytes, and KCs of a mice model of liver disease, and this upregulation contributes to TNF-α production [46]. Furthermore, miR-155 is upregulated in patients with NASH, possibly representing a specific biomarker for this pathological condition [40]. This finding is corroborated by the data presented here, which showed that the animals exposed to PCB 126 developed NASH.

Additionally, increased expression of miR-34a was observed despite the unaltered expression of miR-122 in the liver of the PCB group. miR-34a is upregulated in the liver of patients and rodents with NASH, and its contribution to the development of this disorder appears to occur through mechanisms involving PPARα and SIRT1, which are specific targets of mir-34a [40, 44, 45, 47, 48]. In this context, the upregulation of miR-34a results in the downregulation of hepatic PPARα and SIRT1, which results in reduced fatty acid oxidation and the development of steatosis [44]. These data are in agreement with the results observed in the present study because upregulation of miR-34a and reduced *Ppara* expression were observed in the PCB group. However, an unchanged expression of *Sirt1* suggests that PCB 126 promoted the development of NASH, at least in part, through mechanisms involving reduced fatty acid oxidation in the liver owing to reduced PPARα expression, possibly induced by miR-34a upregulation.

In summary, the present study showed that PCB 126 induced NASH through increased inflammatory processes, impairment of lipid oxidation, and development of hepatic steatosis associated with increased expression of proinflammatory cytokines, reduced expression of PPARα, and upregulation of miR-155 and miR-34a in the liver. To our knowledge, this is the first study to demonstrate the upregulation of miR-155 and miR-34a induced by PCB 126 and their possible contribution to the development of NAFLD. These findings may contribute to the development of novel diagnostic markers and therapeutic strategies.

## Supporting information

**S1 Data. Data from experiments.**
(XLSX)

## Author Contributions

**Conceptualization:** Julio Beltrame Daleprane.

**Data curation:** Fernanda Torres Quitete, Thamara Cherem Peixoto, Bruna Cadete Martins, Angela de Castro Resende, Fabiane Martins, Julio Beltrame Daleprane.

**Formal analysis:** Fernanda Torres Quitete, Ananda Vitória Silva Teixeira, Thamara Cherem Peixoto, Bruna Cadete Martins, Geórgia Correa Atella, Angela de Castro Resende, Daniela de Barros Mucci, Fabiane Martins, Julio Beltrame Daleprane.

**Methodology:** Ananda Vitória Silva Teixeira.

**Writing – original draft:** Julio Beltrame Daleprane.

**Writing – review & editing:** Daniela de Barros Mucci, Julio Beltrame Daleprane.

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
