## [Decision Letter · Decision Letter 0]

26 Mar 2024

PONE-D-24-04954Long-term polychlorinated biphenyl 126 exposure induces liver fibrosis by upregulation of miR-155 and miR-34a in C57BL/6 micePLOS ONE

Dear Dr. Daleprane,

Thank you for submitting your manuscript to PLOS ONE. After careful consideration, we feel that it has merit but does not fully meet PLOS ONE’s publication criteria as it currently stands. Therefore, we invite you to submit a revised version of the manuscript that addresses the points raised during the review process.

Your manuscript was reviewed by two experts and both of them suggested few major comment , which must be addressed during revision. 

We look forward to receiving your revised manuscript.

Kind regards,

Partha Mukhopadhyay, Ph.D.

Section Editor

PLOS ONE

2. To comply with PLOS ONE submissions requirements, in your Methods section, please provide additional information regarding the experiments involving animals and ensure you have included details on methods of sacrifice.

https://linkinghub.elsevier.com/retrieve/pii/S0009279721000053

https://pubs.acs.org/doi/10.1021/acs.jafc.2c01461

https://www.bioseek.eu/in/eng/research/pubmed/Hepatocyte_specific_deletion_of_SIRT1_alters_fatty_acid_metabolism_and_results_in_hepatic_steatosis_and_inflammation_19356714

https://niehs.nih.gov/research/atniehs/labs/lst/mammalian/docs/Purushotham%20et%20al.pdf

In your revision ensure you cite all your sources (including your own works), and quote or rephrase any duplicated text outside the methods section. Further consideration is dependent on these concerns being addressed.

5. PLOS requires an ORCID iD for the corresponding author in Editorial Manager on papers submitted after December 6th, 2016. Please ensure that you have an ORCID iD and that it is validated in Editorial Manager. To do this, go to ‘Update my Information’ (in the upper left-hand corner of the main menu), and click on the Fetch/Validate link next to the ORCID field. This will take you to the ORCID site and allow you to create a new iD or authenticate a pre-existing iD in Editorial Manager. Please see the following video for instructions on linking an ORCID iD to your Editorial Manager account: https://www.youtube.com/watch?v=_xcclfuvtxQ.

6. We note that Figure 1A in your submission contain copyrighted images. All PLOS content is published under the Creative Commons Attribution License (CC BY 4.0), which means that the manuscript, images, and Supporting Information files will be freely available online, and any third party is permitted to access, download, copy, distribute, and use these materials in any way, even commercially, with proper attribution. For more information, see our copyright guidelines: http://journals.plos.org/plosone/s/licenses-and-copyright.

1. You may seek permission from the original copyright holder of Figure 1A to publish the content specifically under the CC BY 4.0 license.

7. We note that the grant information you provided in the ‘Funding Information’ and ‘Financial Disclosure’ sections do not match.

Reviewers' comments:

Reviewer's Responses to Questions

**Comments to the Author**

1. Is the manuscript technically sound, and do the data support the conclusions?

Reviewer #1: Partly

Reviewer #2: Yes

2. Has the statistical analysis been performed appropriately and rigorously? 

Reviewer #1: Yes

Reviewer #2: Yes

3. Have the authors made all data underlying the findings in their manuscript fully available?

Reviewer #1: Yes

Reviewer #2: No

4. Is the manuscript presented in an intelligible fashion and written in standard English?

Reviewer #1: Yes

Reviewer #2: Yes

5. Review Comments to the Author

Reviewer #1: The present study "Long-term polychlorinated biphenyl 126 exposure induces liver fibrosis by upregulation of miR-155 and miR-34a in C57BL/6 mice" analyzed some mechanism of fatty liver disease induced by PCB126. Here are some major concerns:

1. microRNA is not epigenetic marker. The authors need to rewrite the manuscript with accurate concepts.

2. Fibrosis is not an indicator of NASH. If the authors want to prove NASH, infiltration of multiple types of immune cells needs to be experimentally confirmed. Also, the authors need to use experiments to prove hepatocyte death.

3. What is the mechanism to regulate miR-155 and miR-34a expression? How is it related to AHR signal?

4. Which cells in the liver express AHR, miR-155 and miR-34a?

5. The authors need to use molecular biology methods to prove whether TNF-alpha and IL-6 are direct targets of miR-155. And whether PPARalpha is a direct target of miR-34a.

6. To prove miR-155 and miR-34a participate in the pathogenesis, the authors need to use molecular methods to interfere with the expression of these two microRNAs in the liver and repeat the animal experiment, in comparison to the unrelated interference treatment.

7. Fibrosis is only supported by Sirus Red staining, which is too limited. The authors need to perform hydroxyproline analysis, collagen quantification, fibrosis related gene expression to confirm fibrosis.

8. What is the mechanism for cholesterol elevation in PCB treated mice?

Reviewer #2: The authors tested a commonly used pollutant, PCB-126, and observed liver fibrosis and steatosis in mice, as has been shown before. Nevertheless, they also showed increased expression of some miRNAs related to lipid metabolism.

Overall, the results seem concise and nicely presented, and the text is well written, although there are some typos and some mistakes that could need to be revised. Some points of the text are redundant as well, but I’ll bring it along to my revision. Regarding Picrosirius Red, did you make any kind of measurement of the area affected? I would recommend analyzing the area with ImageJ, which is available for free, and several macros could help with the analysis. This could be enriching for the data you show.

The discussion section feels hard to follow. Try to reorganize it to group the results by “themes”. E.g., the Second paragraph (starting in line 245) could come after the third one. Also, this second paragraph gets very redundant with the information that’s about to be presented. Please, reorganize, and check and rewrite what is needed.

Statistical-wise, was there any normality test used? It is important to determine if the data was properly analyzed by the Student’s T-test. Also, something that could add to your conclusions, is correlation. Use simple linear correlation to compare the miRNA you measured with other results you observed. E.g., in Figure 3, you add two correlation analysis between IL-6 and TNF-α by the expression of miR-155, and in Figure 4, the same thing with PPAR-α and miR-34a. Other possibilities are evaluating the correlation between the results you got in the stereology of the liver and the expression of these two miR-34a. I understand that, although correlations could be misleading, once you already are exposing the relation in the discussion section that would be justified. Feel free to try other correlations that could work well and present some of them. This could reinforce your results.

- Line 76: …to date, no study…

- Lines 91 to 97: Information is shown twice. Make the last sentences of the paragraph more complete and you can’t need the subsequent paragraphs.

- Line 100: You can use graphical accents on the words that have it originally, as in São Paulo.

- Line 104: Which drug did you use for euthanasia, pentobarbital or thiopental? Also, use the right symbols {[()]}

- Line 122: Reference 20 doesn’t make sense with what is in the text. It’s about chow, not histology.

- Lines 170 to 186: Gene names and transcripts are two different things. The gene for TNF-α is Tnfa, and, as this is a rodent gene, the first letter is capitalized and subsequent in lowercase, and italics. For it to be clearer, you can add the proper name of the genes in the parenthesis, before the assay ID. Please, be aware of this and fix it along with the text.

- Line 186: Use the proper reference for Livak and Schimittgen.

- Paragraph starting in line 221: Text is hard to follow. Please, rewrite it to present the data with better flow.

- Line 282: Reference of the “other study” should come by the end of the first sentence.

- Line 342: Please, double-check the verbal tense of the verb induce here.

- Line 346: relation instead of relationship.

6. PLOS authors have the option to publish the peer review history of their article (what does this mean?). If published, this will include your full peer review and any attached files.

Reviewer #1: No

Reviewer #2: No

---

## [Author Response · Author response to Decision Letter 0]

10 May 2024

Dear reviewers,

We greatly appreciate your contribution to improving our article. We have addressed each question point by point and hope that all raised concerns have been adequately answered.

We have worked diligently, returning to the laboratory bench to conduct additional experiments aimed at addressing and working through all the issues raised.

We are available for further clarification if needed.

Sincerely,

Julio Daleprane

RESPONSE TO REVIWERS

Reviewer #1: The present study "Long-term polychlorinated biphenyl 126 exposure induces liver fibrosis by upregulation of miR-155 and miR-34a in C57BL/6 mice" analyzed some mechanism of fatty liver disease induced by PCB126. Here are some major concerns:

1. microRNA is not epigenetic marker. The authors need to rewrite the manuscript with accurate concepts.

A: We conducted a thorough reading of the entire manuscript and reformulated the sentences. 

2. Fibrosis is not an indicator of NASH. If the authors want to prove NASH, infiltration of multiple types of immune cells needs to be experimentally confirmed. Also, the authors need to use experiments to prove hepatocyte death.

A: In response to the reviewer's query regarding our methods to elucidate the development of steatosis into fibrosis, we conducted gene expression assays for markers of different cell types. These included MCP1 for macrophages, CXCL1, and IL-8—key chemokines for neutrophil chemotaxis. Notably, CXCL1 was found to significantly elevate neutrophil infiltration (Figure 2). Additionally, beyond the fibrosis previously demonstrated by picrosirius red staining, we quantified the area occupied by the staining, measured tissue hydroxyproline concentrations, and assessed the expression of pro-fibrotic genes such as TGF-beta1, Smad3, Collagen1, and CXCL9. Cell death was evaluated by assessing Caspase3 expression in hepatic tissue (Figure 3).

Figure 2.

Figure 3. 

3. What is the mechanism to regulate miR-155 and miR-34a expression? How is it related to AHR signal?

A: The mechanism by which PCB-126 (polychlorinated biphenyl-126) regulates miR-155 and miR-34a expression is closely tied to its interaction with the aryl hydrocarbon receptor (AhR) signaling pathway. PCB-126 is a potent ligand for AhR, a transcription factor that is activated upon binding with various ligands, including environmental pollutants like dioxins and PCBs. When PCB-126 binds to AHR, it triggers the translocation of the AhR-ligand complex into the nucleus. 

Once inside the nucleus, the AhR-ligand complex can bind to specific DNA sequences known as xenobiotic response elements (XREs) located in the promoter regions of various genes, including those encoding for miRNAs like miR-155 and miR-34a. This binding can either upregulate or downregulate the transcription of these miRNAs, depending on the context and the specific interactions at the gene promoters. miR-155 and miR-34a are microRNAs that play crucial roles in various biological processes, including inflammation, cell cycle regulation, and apoptosis. The regulation of these miRNAs by AhR signaling can influence these processes, potentially leading to various biological effects. The alteration of miR-155 and miR-34a expression by PCB-126 through AhR signaling can have significant pathophysiological implications. For example, dysregulation of these miRNAs may contribute to the development of diseases like cancer, where they are known to play roles in regulating apoptosis, proliferation, and the immune response. In summary, PCB-126 regulates miR-155 and miR-34a expression primarily through its interaction with AhR, leading to changes in gene transcription that can have broad biological and health implications. This pathway is a critical aspect of understanding the toxicological effects of PCBs and their impact on cellular and molecular processes.

4. Which cells in the liver express AHR, miR-155 and miR-34a?

A: As described by Yan et al (2019), AhR is expressed in hepatic stellate cells, hepatocytes and Kupffer cells [1]. It has been described that miR-155 is highly expressed in hepatocytes and Kupffer cells, being involved in inflammatory processes that control innate and adaptive immunity in nonalcoholic fatty liver disease, in addition to an important role in the initial hepatic lipid accumulation and modulating lipid metabolism [2]. miR-34a is expressed in hepatocytes and hepatic stellate cells and was associated to induction of hepatocytic apoptosis, regulation of lipid metabolism and stimulation of hepatic stellate cell activation and fibrosis [3].

[1] Yan J, Tung HC, Li S, Niu Y, Garbacz WG, Lu P, Bi Y, Li Y, He J, Xu M, Ren S, Monga SP, Schwabe RF, Yang D, Xie W. Aryl Hydrocarbon Receptor Signaling Prevents Activation of Hepatic Stellate Cells and Liver Fibrogenesis in Mice. Gastroenterology. 2019 Sep;157(3):793-806.e14. doi: 10.1053/j.gastro.2019.05.066. Epub 2019 Jun 3.

[2] Dongiovanni P, Meroni M, Longo M, Fargion S, Fracanzani AL. miRNA Signature in NAFLD: A Turning Point for a Non-Invasive Diagnosis. Int J Mol Sci. 2018 Dec 10;19(12):3966. doi: 10.3390/ijms19123966.

[3] Hochreuter MY, Dall M, Treebak JT, Barrès R. MicroRNAs in non-alcoholic fatty liver disease: Progress and perspectives. Mol Metab. 2022 Nov;65:101581. doi: 10.1016/j.molmet.2022.101581.

5. The authors need to use molecular biology methods to prove whether TNF-alpha and IL-6 are direct targets of miR-155. And whether PPARalpha is a direct target of miR-34a.

A: Unfortunately, it was not possible to perform molecular biology analyzes to prove such associations, but we revised and rewrote the discussion section so that a direct relationship between these markers is not suggested. 

“However, these associations have already been well described in previous studies. Tili et al (2007) suggest that miR-155 may act directly or indirectly to increase the rate of translation of TNF- transcripts, possibly through their redistribution across the cytoplasm and/or enhancement of TNF- transcript stability. Furthermore, they showed that miR-155 targets transcripts encoding proteins, such as IKKε (IκB kinase ε), FADD (Fas-associated death domain protein) and Ripk1 (receptor (TNFR superfamily)-interacting serine-threonine kinase 1) whose ultimate function results in the activation of the LPS/TNF- pathway, while it increases TNF- production [1]. In addition, other study described that miR-155 is a master regulator of inflammation, enhancing the translation of TNFα during innate immune responses by Toll-like receptor ligands [2]. 

Thus it has already been described that the overexpression of miR-155 promoted activation of NF-κB, through mechanisms involving the downregulation of SOCS1, and elevating the production of proinflammatory cytokines, such TNF-α and IL-6 [3, 4]

Concerning to PPARalpha and its relation to mir-34a, previous studies have already reported that the upregulation of miR-34a resulted in the downregulation of hepatic PPARα and SIRT1 that are the direct targets of miR-34a and that silencing of miR-34a restored the expression of SIRT1 and PPARα, resulting in activation of PPARα downstream genes, such as AMPK and Hydroxymethylglutaryl-CoA Reductase (HMGCR) [2, 5].” 

[1] Tili E, Michaille JJ, Cimino A, Costinean S, Dumitru CD, Adair B, Fabbri M, Alder H, Liu CG, Calin GA, Croce CM. Modulation of miR-155 and miR-125b levels following lipopolysaccharide/TNF-alpha stimulation and their possible roles in regulating the response to endotoxin shock. J Immunol. 2007 Oct 15;179(8):5082-9. doi: 10.4049/jimmunol.179.8.5082.

[2] Dongiovanni P, Meroni M, Longo M, Fargion S, Fracanzani AL. miRNA Signature in NAFLD: A Turning Point for a Non-Invasive Diagnosis. Int J Mol Sci. 2018 Dec 10;19(12):3966. doi: 10.3390/ijms19123966. 

[3] Chen C, Luo F, Liu X, Lu L, Xu H, Yang Q, Xue J, Shi L, Li J, Zhang A, Liu Q. NF-kB-regulated exosomal miR-155 promotes the inflammation associated with arsenite carcinogenesis. Cancer Lett. 2017 Mar 1;388:21-33. doi: 10.1016/j.canlet.2016.11.027.

[4] Tan L, Jiang W, Lu A, Cai H, Kong L. miR-155 Aggravates Liver Ischemia/reperfusion Injury by Suppressing SOCS1 in Mice. Transplant Proc. 2018 Dec;50(10):3831-3839. doi: 10.1016/j.transproceed.2018.08.060. 

[5] Ding J, Li M, Wan X, Jin X, Chen S, Yu C, Li Y. Effect of miR-34a in regulating steatosis by targeting PPARα expression in nonalcoholic fatty liver disease. Sci Rep. 2015 Sep 2;5:13729. doi: 10.1038/srep13729.

6. To prove miR-155 and miR-34a participate in the pathogenesis, the authors need to use molecular methods to interfere with the expression of these two microRNAs in the liver and repeat the animal experiment, in comparison to the unrelated interference treatment.

A: Unfortunately, we were unable to perform these experiments, but we revised and rewrote the discussion section to try to minimize this direct association of miRs with the pathogenesis of NASH.

7. Fibrosis is only supported by Sirus Red staining, which is too limited. The authors need to perform hydroxyproline analysis, collagen quantification, fibrosis related gene expression to confirm fibrosis.

A: We performed additional experiments as suggested (Fig 3). 

8. What is the mechanism for cholesterol elevation in PCB treated mice?

A: PCB-126 is known to be a potent endocrine disruptor. It can interfere with thyroid and steroid hormone receptors, thereby affecting lipid metabolism in the liver. Changes in hormone levels may result in increased cholesterol synthesis or reduced elimination. Simultaneously, exposure to PCB-126 can alter liver metabolic pathways, increasing cholesterol biosynthesis and/or decreasing its conversion into bile acids. This may be caused by changes in the expression or activity of key enzymes such as HMG-CoA reductase, which is crucial in the cholesterol biosynthetic pathway. Additionally, PCB-126 can induce oxidative stress and inflammation in the liver. Oxidative stress may damage liver cells, while inflammation can trigger a lipid response that includes increased cholesterol synthesis as a defense mechanism. Finally, PCB-126 may affect the function and expression of lipid transporter proteins, such as low-density (LDL) and high-density (HDL) lipoproteins, impacting cholesterol transport and removal from the liver. To improve the quality of the work, we have included the information in the text. [1,2]

1. Mohammadparast-Tabas P, Arab-Zozani M, Naseri K, Darroudi M, Aramjoo H, Ahmadian H, et al. Polychlorinated biphenyls and thyroid function: a scoping review. Rev Environ Health. 2023. doi:10.1515/reveh-2022-0156

2. Yang Y, Mei G, Yang L, Luo T, Wu R, Peng S, et al. PCB126 impairs human sperm functions by affecting post-translational modifications and mitochondrial functions. Chemosphere. 2024;346: 140532. doi:10.1016/j.chemosphere.2023.140532

 

Reviewer #2: The authors tested a commonly used pollutant, PCB-126, and observed liver fibrosis and steatosis in mice, as has been shown before. Nevertheless, they also showed increased expression of some miRNAs related to lipid metabolism.

Overall, the results seem concise and nicely presented, and the text is well written, although there are some typos and some mistakes that could need to be revised. Some points of the text are redundant as well, but I’ll bring it along to my revision. 

1.Regarding Picrosirius Red, did you make any kind of measurement of the area affected? I would recommend analyzing the area with ImageJ, which is available for free, and several macros could help with the analysis. This could be enriching for the data you show.

A: We performed additional experiments as suggested (Fig 3). 

2.The discussion section feels hard to follow. Try to reorganize it to group the results by “themes”. E.g., the Second paragraph (starting in line 245) could come after the third one. Also, this second paragraph gets very redundant with the information that’s about to be presented. Please, reorganize, and check and rewrite what is needed.

A: The discussion section was reviewed and we reorganize and rewrite some paragraphs to make it easier to understand.

3.Statistical-wise, was there any normality test used? It is important to determine if the data was properly analyzed by the Student’s T-test. Also, something that could add to your conclusions, is correlation. Use simple linear correlation to compare the miRNA you measured with other results you observed. E.g., in Figure 3, you add two correlation analysis between IL-6 and TNF-α by the expression of miR-155, and in Figure 4, the same thing with PPAR-α and miR-34a. Other possibilities are evaluating the correlation between the results you got in the stereology of the liver and the expression of these two miR-34a. I understand that, although correlations could be misleading, once you already are exposing the relation in the discussion section that would be justified. Feel free to try other correlations that could work well and present some of them. This could reinforce your results.

A: Dear Reviewer, Thank you for the opportunity to clarify the conduct of our data analyses. To carry out the statistical analyses, we initially performed a normality test on all data, as now described in the methodology section. Subsequent statistical analyses were then performed. We attempted to conduct a correlation test with the studied variables; however, unfortunately, the sample size was too small to establish this relationship. As a result, we changed how we present the results, and no longer claim causality between miRNAs and other inflammatory and fibrotic indicators. We now merely discuss these in light of existing data in the literature.

4.Line 76: …to date, no study…

A: Done.

5.Lines 91 to 97: Information is shown twice. Make the last sentences of the paragraph more complete and you can’t need the subsequent paragraphs.

A: We rewrite the sentence.

6.Line 100: You can use graphical accents on the words that have it originally, as in São Paulo.

A: Done.

7. Line 104: Which drug did you use for euthanasia, pentobarbital or thiopental? Also, use the right symbols {[()]}

A: Sorry for this mistake. The drug used was sodium thiopental and the sentence has already been corrected.

8. Line 122: Reference 20 doesn’t make sense with what is in the text. It’s about chow, not histology.

A:It has been corrected.

9. Lines 170 to 186: Gene names and transcripts are two different things. The gene for TNF-α is Tnfa, and, as this is a rodent gene, the first letter is capitalized and subsequent in lowercase, and italics. For it to be clearer, you can add the proper name of the genes in the parenthesis, before the assay ID. Please, be aware of this and fix it along with the text.

A: We correct this throughout the text.

10. Line 186: Use the proper reference for Livak and Schimittgen.

A: Done.

11. Paragraph starting in line 221: Text is hard to follow. Please, rewrite it to present the data with better flow.

A: We rewrote the paragraph to improve understanding of the data presented.

12. Line 282: Reference of the “other study” should come by the end of the first sentence.

A: The reference was included at the end of the sentence.

13. Line 342: Please, double-check the verbal tense of the verb induce here.

A: Done.

14. Line 346: relation instead of relationship.

A: Done.

---

## [Decision Letter · Decision Letter 1]

16 Jun 2024

PONE-D-24-04954R1Long-term exposure to polychlorinated biphenyl 126 induces liver fibrosis and upregulates miR-155 and miR-34a in C57BL/6 micePLOS ONE

Dear Dr. Daleprane,

Thank you for submitting your manuscript to PLOS ONE. After careful consideration, we feel that it has merit but does not fully meet PLOS ONE’s publication criteria as it currently stands. Therefore, we invite you to submit a revised version of the manuscript that addresses the points raised during the review process.

Your manuscript was reviewed by same experts and one of them raised few comments , which must be addressed.

We look forward to receiving your revised manuscript.

Kind regards,

Partha Mukhopadhyay, Ph.D.

Section Editor

PLOS ONE

Journal Requirements:

Reviewers' comments:

Reviewer's Responses to Questions

**Comments to the Author**

1. If the authors have adequately addressed your comments raised in a previous round of review and you feel that this manuscript is now acceptable for publication, you may indicate that here to bypass the “Comments to the Author” section, enter your conflict of interest statement in the “Confidential to Editor” section, and submit your "Accept" recommendation.

Reviewer #1: (No Response)

Reviewer #2: All comments have been addressed

2. Is the manuscript technically sound, and do the data support the conclusions?

Reviewer #1: Partly

Reviewer #2: Yes

3. Has the statistical analysis been performed appropriately and rigorously? 

Reviewer #1: Yes

Reviewer #2: Yes

4. Have the authors made all data underlying the findings in their manuscript fully available?

Reviewer #1: Yes

Reviewer #2: Yes

5. Is the manuscript presented in an intelligible fashion and written in standard English?

Reviewer #1: Yes

Reviewer #2: Yes

6. Review Comments to the Author

Reviewer #1: The current version of "Long-term exposure to polychlorinated biphenyl 126 induces liver fibrosis and upregulates miR-155 and miR-34a in C57BL/6 mice" made significant improvement compared to the original submission. Here are some more major and minor concerns:

Major concern

1. Caspase-3 expression cannot indicate apoptosis. The authors should either perform western blot to evaluate cleaved caspase-3, or IHC for TUNEL in histological samples. If using western blot to detect cleaved caspase-3, total or full-length caspase-3 protein level should also be shown still.

Minor concern

1. IL-8 is typically not considered present in murine. Human IL-8 equivalent in mice is CXCL1, which the authors have already examined. IL-8 measurement can be deleted.

2. miR-155 expression level should be moved to figure 4 from figure 2

3. The authors answered the potential mechanism linking AHR to miR-155 and miR-34a regulation in the rebuttal letter. Please include this information to the discussion section with proper citations.

Reviewer #2: I am satisfied with the alterations the authors made to the manuscript. I recommend its publication.

7. PLOS authors have the option to publish the peer review history of their article (what does this mean?). If published, this will include your full peer review and any attached files.

Reviewer #1: No

Reviewer #2: No

---

## [Author Response · Author response to Decision Letter 1]

19 Jun 2024

Dear Reviewer,

We would like to express our gratitude for the thorough review and valuable feedback on our manuscript. We have carefully considered each of your comments and have made the necessary revisions accordingly. Below, we address each of your concerns in detail:

Major Concern

1. Caspase-3 Expression:

 We acknowledge that caspase-3 expression alone cannot definitively indicate apoptosis. In response to your suggestion, we have performed a Western blot analysis to evaluate cleaved caspase-3. Additionally, we have included the total or full-length caspase-3 protein level in our results. The updated data can be found in the revised manuscript (Figure 3I and 3J). We believe this provides a more comprehensive evaluation of apoptosis in our samples.

Minor Concerns:

1. IL-8 Measurement:

 We appreciate your observation regarding IL-8. As suggested, we have removed the IL-8 measurement from our manuscript, considering that IL-8 is not typically present in murine models and we have already examined the murine equivalent, CXCL1. The manuscript has been updated accordingly to reflect this change.

2. miR-155 Expression Level:

 We have moved the miR-155 expression data from Figure 2 to Figure 4 as requested. This rearrangement aligns better with the overall flow of our results and discussion.

3. Mechanism Linking AHR to miR-155 and miR-34a Regulation:

 We have incorporated the information regarding the potential mechanism linking AHR to the regulation of miR-155 and miR-34a into the discussion section of our manuscript. Proper citations have been added to support this information. We believe this addition strengthens the discussion and provides a clearer understanding of the mechanisms involved.

We sincerely thank you again for your constructive feedback, which has significantly improved the quality of our manuscript. We hope that the revised version meets your expectations and addresses all your concerns satisfactorily.

Best regards,

Julio Daleprane

---

## [Decision Letter · Decision Letter 2]

23 Jul 2024

Long-term exposure to polychlorinated biphenyl 126 induces liver fibrosis and upregulates miR-155 and miR-34a in C57BL/6 mice

PONE-D-24-04954R2

Dear Dr. Daleprane,

We’re pleased to inform you that your manuscript has been judged scientifically suitable for publication and will be formally accepted for publication once it meets all outstanding technical requirements.

Kind regards,

Partha Mukhopadhyay, Ph.D.

Section Editor

PLOS ONE

Additional Editor Comments (optional):

Reviewers' comments:

Reviewer's Responses to Questions

**Comments to the Author**

1. If the authors have adequately addressed your comments raised in a previous round of review and you feel that this manuscript is now acceptable for publication, you may indicate that here to bypass the “Comments to the Author” section, enter your conflict of interest statement in the “Confidential to Editor” section, and submit your "Accept" recommendation.

Reviewer #1: All comments have been addressed

2. Is the manuscript technically sound, and do the data support the conclusions?

Reviewer #1: (No Response)

3. Has the statistical analysis been performed appropriately and rigorously? 

Reviewer #1: (No Response)

4. Have the authors made all data underlying the findings in their manuscript fully available?

Reviewer #1: (No Response)

5. Is the manuscript presented in an intelligible fashion and written in standard English?

Reviewer #1: (No Response)

6. Review Comments to the Author

Reviewer #1: (No Response)

7. PLOS authors have the option to publish the peer review history of their article (what does this mean?). If published, this will include your full peer review and any attached files.

Reviewer #1: No

---

## [Editor Report · Acceptance letter]

1 Aug 2024

PONE-D-24-04954R2 

PLOS ONE

Dear Dr. Daleprane, 

I'm pleased to inform you that your manuscript has been deemed suitable for publication in PLOS ONE. Congratulations! Your manuscript is now being handed over to our production team.

Kind regards, 

on behalf of

Dr. Partha Mukhopadhyay 

Section Editor

PLOS ONE